# Rhizobia Contribute to Salinity Tolerance in Common Beans (*Phaseolus vulgaris* L.)

**DOI:** 10.3390/cells11223628

**Published:** 2022-11-16

**Authors:** Clabe Wekesa, George O. Asudi, Patrick Okoth, Michael Reichelt, John O. Muoma, Alexandra C. U. Furch, Ralf Oelmüller

**Affiliations:** 1Matthias Schleiden Institute of Genetics, Bioinformatics and Molecular Botany, Friedrich-Schiller-University Jena, Dornburger Str. 159, 07743 Jena, Germany; 2Department of Biochemistry, Max Planck Institute for Chemical Ecology, Hans-Knöll-Str. 8, 07745 Jena, Germany; 3Department of Biochemistry, Microbiology and Biotechnology, Kenyatta University, P.O. Box 43844, Nairobi 00100, Kenya; 4Department of Biological Sciences, Masinde Muliro University of Science and Technology, P.O. Box 190-50100, Kakamega 50100, Kenya

**Keywords:** rhizobia, RNA_Seq, transcriptomics, common beans, salinity tolerance, GABA, osmotolerance, meta-RNA-Seq

## Abstract

Rhizobia are soil bacteria that induce nodule formation on leguminous plants. In the nodules, they reduce dinitrogen to ammonium that can be utilized by plants. Besides nitrogen fixation, rhizobia have other symbiotic functions in plants including phosphorus and iron mobilization and protection of the plants against various abiotic stresses including salinity. Worldwide, about 20% of cultivable and 33% of irrigation land is saline, and it is estimated that around 50% of the arable land will be saline by 2050. Salinity inhibits plant growth and development, results in senescence, and ultimately plant death. The purpose of this study was to investigate how rhizobia, isolated from Kenyan soils, relieve common beans from salinity stress. The yield loss of common bean plants, which were either not inoculated or inoculated with the commercial *R. tropici* rhizobia CIAT899 was reduced by 73% when the plants were exposed to 300 mM NaCl, while only 60% yield loss was observed after inoculation with a novel indigenous isolate from Kenyan soil, named S3. Expression profiles showed that genes involved in the transport of mineral ions (such as K^+^, Ca^2+^, Fe^3+^, PO_4_^3−^, and NO_3_^−^) to the host plant, and for the synthesis and transport of osmotolerance molecules (soluble carbohydrates, amino acids, and nucleotides) are highly expressed in S3 bacteroids during salt stress than in the controls. Furthermore, genes for the synthesis and transport of glutathione and γ-aminobutyric acid were upregulated in salt-stressed and S3-inocculated common bean plants. We conclude that microbial osmolytes, mineral ions, and antioxidant molecules from rhizobia enhance salt tolerance in common beans.

## 1. Introduction

Various environmental stresses such as floods, drought, and salinity continue to undermine the cultivation and growth of crops. Soil salinity is one of the main threats in agriculture, its productivity, and quality of crops [1,2]. According to Shrivastava and Kumar [2], the soil is defined as saline if the electrical conductivity (EC) of its rhizosphere’s saturation extract (EC_e_) exceeds 4 dS m^−1^, i.e., approximately 40 mM NaCl with 15% exchangeable sodium at 25 °C. Worldwide, about 20% of cultivable and 33% of irrigation land is saline, and it is estimated that around 50% of the arable land will be saline by 2050 [3]. High surface evaporation, irrigation with saline water, low precipitation, weathering of native rocks, and poor cultural practices are the main reasons for the approximately 10% yearly increase in worldwide salinization [2].

Adverse effects of salinity on crops are mainly osmotic toxicities and the uptake of ions in toxic concentrations [4]. Increased osmotic values reduce the soil solution’s water potential which restricts water and nutrient uptake by the roots. Excessive accumulation of Na^+^, Cl^−^, HCO_3_^−^, CO_3_^2−^ and other ions leads to metabolic imbalances or even toxicity for the plants [5]. Furthermore, elevated levels of carbonates/bicarbonates in the soil cause reduced solubility of essential elements for plant growth like iron (Fe) and phosphorus (P) or the trace elements zinc (Zn) and manganese (Mn), through precipitation [6]. For instance, high levels of bicarbonates induced Fe deficiency and this caused chlorosis in sorghum and maize [7] and Zn deficiency in rice resulting is discolored leaves and stunted growth [8]. Salinity causes also nutrient deficiencies or imbalances through competition between Na^+^ and Cl^−^ ions with important ions such as K^+^, Ca^2+^, and NO_3_^−^ [9,10]. Through clay dispersion, Na^+^ impedes penetration of the roots into the rhizosphere which ultimately reduces plant growth. Clay dispersion also leads to poor aeration, which further promotes uptake of Na^+^ and Cl^−^ and their transport to the shoots [11]. Sodium may also accumulate in the leaf apoplasm leading to turgor loss, dehydration, and death of leaf cells and tissues [12].

Abscisic acid synthesized during salt stress causes stomatal closure by inhibition of guard cell expansion, which results in photoinhibition and oxidative stress due to reduced photosynthesis rates [13]. Sodium has a strong inhibitory effect on K^+^ uptake by the roots, therefore, plants growing on soils with excess Na^+^ ions often suffer under K^+^ deficits [14]. Potassium is critical for cell turgor, membrane potential, and various enzyme activities [15,16]. In particular, a high Na^+^/K^+^ ratio in the cytosol inhibits cytosolic enzymatic activities. However, K^+^ uptake and the K^+^/Na^+^ balance in Na^+^-stressed plants is partially rescued by a simultaneously occurring increased uptake of Ca^2+^ [17]. Calcium activates intracellular signaling pathways which regulate the expression and activity of Na^+^ and K^+^ transporters and suppress Na^+^ import by nonselective cation channels [18]. Salinity also inhibits reproductive development by inhibiting microsporogenesis and stamen filament elongation, enhanced programmed cell death, senescence of fertilized embryos, and ovule abortion [2,19].

Common bean (*Phaseolus vulgaris* L.) is the daily diet of more than 300 million people worldwide. This crop is the world’s most important food legume, far more than chickpeas, lentils, faba bean, and cowpea [20]. Nutritionists classify common bean as a nearly perfect food because of its high protein content and a generous amount of fiber, complex carbohydrates, and other dietary necessities [21]. However, the crop is sensitive to salinity and suffers yield losses under soil salinities of less than 2 dSm^−1^ [22]. Salt stress increased the germination time [23] and inhibited nodulation [24]. As a consequence, nitrogen fixation of the bacteroids is decreased due to their lower respiratory capacity and growth in common bean roots [25,26]. The number of leaves, plant height, root and shoot weight, transpiration, photosynthesis rate, and stomatal conductance were adversely affected in all common bean cultivars which were studied under salt stress [27]. Ghassemi-Golezani et al. [21] reported a loss of common bean yield up to 19.4% during salinity stress. Salinity decreases nodulation and nitrogen fixation mainly by reducing the activity of nitrogenase enzyme in the nodules of all legumes [28]. Salinity can alter nodule ultrastructure by increasing oxygen diffusion resistance in the nodules [29]. Accompanied by decreasing N2-fixation in nodules under salinity is the activity of reactive oxygen species (ROS)-scavenging enzymes such as catalase, superoxide dismutase, and ascorbate peroxidase and the level of antioxidants like ascorbic acid [30] resulting in accumulation of ROS in the nodules. However, the intensity of this adverse effects on nodule function depends on inoculated rhizobial strain and duration of exposure to saline conditions [31].

Salinity covers most semi-arid, arid, and coastal regions of Kenya. Arid and semi-arid areas in this country are characterized by high salt contaminations which originate primarily from weathering volcanic parent materials. In contrast, the salinity in the soils of the coastal areas originates mainly from the inundation of seawater, resulting in extremely high Na^+^, Cl^−^, HCO_3_^−^ and CO_3_^2−^ concentrations [6]. Salinity occurs on 1.92 million ha agricultural soil, equivalent to 4.26% of the total arid and semi-arid regions in the country [6]. Moreover, the area for which the electrical conductivity of the soil’s saturation paste extract (ECe) exceeds 4 dSm^−1^ is even larger. According to Sombroek et al. [32], these soils are estimated to cover approximately 18.0 million ha, accounting for about 40% of Kenya’s arid and semi-arid soils. These data demonstrate that drought remains the main challenge to agriculture, in particular for common bean cultivation in Kenya’s semi-arid and arid regions. In major agricultural regions of Kenya, the yield of common bean can potentially reach 1200 kg ha^−1^; however, the yield remains below 500 kg ha^−1^ [33,34]. Improved water and soil management can promote agricultural production on soil challenged by salinity, however, until recently, this was not successful in the Kenyan agriculture [35]. Although suitable biotechnological approaches may help to improve crop yield and soil health [36], those techniques are neglectable in comparison to strategies which deal with the more than 50% crop loss due to salinity. Soil bacteria can help plants to better adapt to saline soils with high osmotic stress, poor physical conditions, nutritional disorders and toxicities [2]. In particular, rhizosphere bacteria which provide the host with essential growth promoting compounds, such as N from fixed nitrogen, P from insoluble phosphates, insoluble Fe due to sequestration by bacterial siderophores, or phytohormones, are ideal and natural tools to promote crop productivity [37]. Rhizobia are ideal candidates for common bean production on saline soil because they fix nitrogen, solubilize phosphorus, protect the host against phytopathogens, and enable iron mobility to the plant cells [38,39,40]. Rhizobia-colonized common bean plants also perform better on saline soils, although the mechanism is not known. In this study, we present the novel rhizobium S3 isolated from common bean nodule from Kenya in our earlier study [41]. In this study, which was performed to determine the local isolates’ competence in Kenyan soils, S3 produced in common beans 2.1 times more seed dry weight than the untreated plants. Nodule formation by isolate S3 on common beans treated with the salt was comparable to those formed by the untreated plants S3 (data not shown). Plants inoculated with this rhizobium perform much better under salt stress in comparison to those which were inoculated with the commonly used and commercially available rhizobium strain CIAT899. S3 might be a promising candidate for agricultural applications in Kenyan saline soils.

## 2. Materials and Methods

### 2.1. Rhizobia Exposure to Salinity

Rhizobia strains (B3, S2, and S3), recently isolated from common bean nodules grown in Western Kenya and which showed high symbiotic efficiency and adaptability to various abiotic stresses such as acidity and aluminium toxicity [40,42] were grown in 50 mL minimal media in 250 mL flasks and maintained at 30 °C in an orbital shaker. Additionally, the commercial isolate *Rhizobium tropici* CIAT899, obtained from Nairobi microbial resource centers (MIRCENs), was used alongside the indigenous isolates. The media contained 1000 μM CaCl_2_⋅2H_2_O, 500 μM MgSO_4_⋅7H_2_O, 50 μM KCl, 25 μM FeEDTA, 3000 μM KH_2_PO_4_, 10 μM H_3_BO_3_, 1 μM MnSO_4_⋅H_2_O, 0.5 μM ZnSO_4_⋅7H_2_O, 0.1 μM CuSO_4_⋅5H_2_O, 0.025 μM Na_2_MoO_4_, 0.005 μM CoCl_2_⋅6H_2_O, 1.8 g/L sodium glutamate, and 10 g/L mannitol. Sodium glutamate were supplied as a filter sterilized solution. Six treatments with NaCl were applied; 0.1%, 0.25%, 0.5%, 1.0%, 2.0%, and 5.0%. After 48 h, the optical density of the cells was measured at OD_600_. The number of cells in each treatment was estimated from the standard curve obtained by the plate count method from optical density [43], and the experiments were independently repeated 5 times.

### 2.2. Salinity Exposure to Common Beans and Co-Cultivation with Rhizobia

Common bean seeds (Rosecoco cultivar) of uniform size were surface sterilized with 70% ethanol for 30 s and 5% sodium hypochlorous for 4 min and rinsed in 10 changes of sterile distilled water. The seeds were pre-germinated in sterile vermiculite (Raiffeisen Gartenbau GmbH & Co. KG, Erfurt, Germany) before being transferred to plastic jars supplied with sterile vermiculite. Three days later, the seedlings were inoculated with 1 mL of yeast extract mannitol broth with rhizobia (OD_600_ = 0.5). The experimental set up was as follows: plants supplied with reduced nitrogen (1 mM KNO_3_), plants inoculated with standard rhizobia CIAT899 or those inoculated with the indigenous rhizobia isolate S3. They were regularly moisturized and nourished with nitrogen free nutrient solution [44]. Seven days post-inoculation, half of the plants in each setup were irrigated with the nutrient media containing 300 mM NaCl. The other half (not treated with NaCl) were used as positive control. After 28 days, 3 × 5 nodules were carefully removed from 10 randomly picked plants for three replicates and immediately frozen in liquid nitrogen. Plant shoots were then dried in the oven for 96 h at 70 °C and weighed to obtained the shoot dry weight. The resulting yield from each treatment was calculated as follows:$$\mathrm{Percent}\;\mathrm{Yield}=\frac{\mathrm{SDW}\;\left(t\right)}{\mathrm{SDW}\;\left(u\right)}\times 100$$
whereas SDW (t) is the shoot dry weight of the plants treated with NaCl, while SDW (u) the shoot dry weights of the untreated plants.

### 2.3. RNA Extraction, Sequencing, and Bioinformatics Analysis

RNA was extracted from nodules obtained from plants inoculated with isolate S3 and either treated or untreated with NaCl with three replications in each case. Nodules were crushed into a fine powder in liquid nitrogen followed by total RNA isolation with TRIzol^®^ reagent (Invitrogen, Waltham, MA, USA) following the manufacturer’s protocol [42]. RNA quantity was determined by measuring optical density at 260 nm while quality was estimated from 260/280 nm and 260/230 nm optical density ratios. Only samples with optical density ratios of 260/280 and 260/230 were send for RNA sequencing at Novogene Company Limited (Cambridge, UK) with Illumina Miseq (Illumina, Inc., San Diego, CA, USA) using company established procedures.

We first checked for the quality of reads using FastQC v0.11.8 [45] before removing low quality reads (Phred values below 20) and those containing adaptors using Trimmomatic v0.36 [46]. The resulting sequence reads are a mixture of RNA from rhizobia and common bean nodule tissue (meta-RNA). To utilize only reads that originated from rhizobia, we aligned the reads to the *R. phaseoli* strain R650 (NCBI accession: GCA_001664385.1) genome with Bowtie2 v2.4.4 [47]. Reads overlapping genes were counted using htseq-count from the HTSeq v2.0.2 [48] and differential gene expression was performed in DESeq2 v1.30.1 [49]. Differentially expressed genes (DEGs) were visualized as a volcano plot using Bioinfokit v2.1.0 [50].

Protein sequences belonging to DEGs were extracted from the proteome of *R. phaseoli* (accession GCF_001664385.1) with inhouse bash script and searched locally against the NCBI’s Conserved Domain Database (CDD) [51] using reverse position-specific blast (RPS-BLAST) script from BLAST + v2.13.0 for conserved domains. The domains were then searched against NCBI’s Clusters of Orthologous Groups of proteins (COGs) database [52] using cdd2cog.pl v0.1 [53] for the gene function. DEGs were also mapped to the Kyoto Encyclopedia of Genes and Genomes (KEGG) base [54] for further functional annotation of genes through the KEGG Automatic Annotation Server (KAAS) [55].

### 2.4. Quantitative Real-Time PCR (qPCR)

We performed quantitative PCR for aspartate aminotransferase, glutathione synthase, glutamate decarboxylase 1, ABC transporter permease, and potassium transporter ATPase genes as a validation of RNA-Seq results and to confirm the importance of the corresponding proteins for either synthesis or efflux into the host plants. The house-keeping genes, gyrase A (*gyr*A) and ATPase A (*atp*A) were run together with the five genes for internal normalization. RNA was extracted from the nodules of salt treated and untreated plants using TRIzol^®^ reagent (Invitrogen, Waltham, MA, USA). A cDNA library was prepared from high quality RNA using RevertAid First Strand cDNA Synthesis Kit (Thermo Fisher Scientific, Waltham, MA, USA) guided by the manufacturer’s protocol. The reaction mixture was as follows: template (10 µg), 2.5 mM dNTPs (2 µL) (Thermo Fisher Scientific, Waltham, MA, USA), 10× DreamTaq buffer (2 µL) (Thermo Fisher Scientific, Waltham, MA, USA), 5 U/µL DreamTaq DNA Polymerase (0.25 µL) (Thermo Fisher Scientific, Waltham, MA, USA), 20× EvaGreen^®^ Dye (BIOZOL Diagnostica Vertrieb GmbH, Eching, Germany), 10 pM primers (1 µL) for each forward and reverse reactions, and water (to 20 µL). Primers used in this study (Appendix A) were designed with the by Primer3Plus webserver [56] with default settings except for adjusting melting temperature to range from 60–65 °C. The real-time qPCR was performed on the CFX Connect Real-Time PCR Detection System (Bio-Rad) (Bio-Rad Laboratories GmbH, Feldkirchen, Germany) with the following thermocycling conditions: initial denaturation 95 °C; 3 min, denaturation 95 °C; 10 s, annealing 60 °C; 50 s, extension 72 °C; 1 min, cycles 40×. Analysis of the resulting data was implemented by delta-delta-Ct (2^^−∆∆CT^) method [57].

### 2.5. Quantification of γ-Aminobutyric Acid (GABA) and Amino Acids

The nodules were extracted in 0.5 mL of methanol and the extract was then diluted in the ratio of 1:10 (*v*:*v*) in water containing U-^13^C, ^15^N labeled amino acid mix (algal amino acids ^13^C, ^15^N, Isotec, Miamisburg, OH, USA) with a concentration of 10 µg of the mix per mL. GABA in the diluted extracts was directly analyzed by LC-MS/MS according to Scholz et al. [58]. The Agilent 1260 HPLC system (Agilent Technologies, Böblingen, Germany) was used to perform chromatography. Separation of 1 µL of the diluted sample was performed on a Zorbax Eclipse XDB-C18 column (50 × 4.6 mm, 1.8 µm, Agilent Technologies, Germany). 0.05% of formic acid in water was used as mobile phase A while acetonitrile was utilized as mobile phase B. The elution profile was as follows: 0–1 min, 3% B in A; 1–2.7 min, 3–100% B in A; 2.7–3 min 100% B, and 3.1–6 min 3% B in A. The mobile phase flow rate was put at 1.1 mL/min while the column temperature was put at 25 °C. The liquid chromatography was coupled to a QTRAP6500 tandem mass spectrometer (AB Sciex, Darmstadt, Germany) that was equipped with a Turbospray ion source operated in a positive ionization mode. The instrument parameters were optimized by the infusion experiments with pure standards. The ion spray voltage was maintained at 5500 eV. The turbo gas temperature was set at 620 °C, nebulizing gas, curtain gas, and heating gas were set at 70 psi, 40 psi, and 70 psi, respectively, while and collision gas was set at medium. The analyte parent ion → product ion: GABA (*m*/*z* 104.1 → 87.1; DP 51, CE 17), U-13C, 15N-Ala (*m*/*z* 94.1 → 47.1; DP 51, CE 17) was then monitored with the multiple reaction monitoring (MRM). Table 1 shows the MRMs of the amino acids proline, valine, leucine, and isoleucine. Both the Q1 and Q3 quadrupoles were kept at unit resolution. The Analyst 1.5 software (AB Sciex, Darmstadt, Germany) was employed in the data acquisition and processing. Each amino acid’s concentration was estimated relative to its corresponding labelled amino acids. GABA in the sample was quantified using U-^13^C, ^15^N-Ala, in which a response factor of 1.0 was applied.

## 3. Results and Discussion

### 3.1. Isolate S3 Is a Salt-Tolerant Rhizobia Strain

We first investigated the tolerance ability of three rhizobial strains (S3, S2 and B2) which were isolated from common bean nodules grown in Kenyan soil to grow in media with various concentrations of aqueous sodium chloride. Standard *R. tropici* CIAT899 was used as a control. The three isolates and the control CIAT899 demonstrated the highest growth rate with 1 mM NaCl in the medium, which dramatically decreased with increasing NaCl concentrations. Isolate S3 exhibited a significantly higher growth rate at 2 and 5 mM NaCl treatments than all other isolates except in the medium with 1 mM NaCl where its growth rate was equivalent to that of B3 (Figure 1). The osmoadaptive mechanisms of rhizobia differ among different strains depending on the osmolality of their periplasmic compartments and depends on their low molecular weight organic solute profiles [59], their extracellular polysaccharide patterns, the cell size and morphology [60]. Determination of tolerance to salinity should therefore be part of pre-screening procedures in the isolation of rhizobia for commercialization purposes.

### 3.2. Isolate S3 Reduced the Effect of Salinity and Weight Loss in Common Beans

Inoculated and un-inoculated common bean plants were exposed to NaCl stress. Irrespective of the inoculation, salinity caused yellowing of leaves of all plants. However, the yellowing phenotype was more severe in uninoculated plants compared to those treated with rhizobia (Figure 2A). Rady et al. [61] showed that accumulation Na^+^ and Cl^−^ ions induced nutritional imbalance by restriction of ion absorption including K^+^, Mg^2+^ and Ca^2+^, and NO_3_^−^ was the main reason for the yellow phenotype in the leaves. Significant reduction of leaf yellowing in rhizobia-treated plants suggests that bacteria might restore the nutritional balance in the host. Furthermore, rhizobia-treated plants showed significantly higher capacities for water uptake compared to the untreated controls (data not shown). This indicates that the bacteria release osmotic stress in the salt-stressed hosts by interfering with the ion homeostasis.

Saline stress caused about 60–75% shoot dry weight loss of uninoculated common bean plants under our growth conditions. S3 inoculated plants lost about 60% yield whereas 73% loss was observed for uninoculated plants and those inoculated with CIAT899 (Figure 2B). Therefore, S3 inoculated plants performed significantly better than those inoculated with CIAT899 or without inoculation. These results indicate that the novel strain S3 but not CIAT899 protects their hosts against salt stress. Similarly, Jebara et al. [62] showed that common bean inoculated with native salt tolerant rhizobia have larger shoots than those inoculated with CIAT899 in hydroponic cultures. Additionally, Estévez et al. [63] found that the beneficial effects of rhizobia on plant growth depend on the bacterial species. Cordovilla et al. [64] postulated that the salt stress-induced decline in the host’s dry weights is caused by lower nitrogen content in shoots. The extend of the stress symptoms for the plants depends on the nitrogen fixation capacity of the rhizobia under salt stress. Thus, salinity tolerance of the rhizobial strains is probably a crucial parameter for the performance of inoculated common beans under salt stress.

### 3.3. Rhizobia Gene Expression in the Nodules during Salinity Stress

Expression profiles for the bacterial genes in the nodules of the plants were obtained for three biological replicates. The transcript abundance for each gene as estimated by bedtools, and the Deseq2-based analysis uncovered that 1069 (Appendix A) bacteroid genes in the nodules were differentially expressed when the plants were exposed to 300 mM salt stress. With a cutoff of adjusted *p*-value ≤ 0.05 and fold change ≥2, 146 genes were up-regulated and 151 genes down-regulated during salt stress (Figure 3). The up-regulated genes were further analyzed because they might enable synthesis or transport of substances relevant for salinity tolerance of the host.

### 3.4. Annotation of the Up-Regulated DEGs

Out of 146 upregulated genes, 135 (92.4%) could be mapped to the cluster of orthologous groups (COGs) (Appendix A) while 100 (68.5%) to the KEGG pathways. The most enriched COGs were those involved in energy production, metabolism and transport of amino acids, carbohydrates, and inorganic ions (Figure 4).

#### 3.4.1. Amino Acid Transport and Metabolism

The gene for a choline dehydrogenase (COG2303), an enzyme that catalyzes the first reaction in the formation of the osmoprotectant glycine betaine through the choline oxidation pathway was upregulated under salt stress. Although accumulation of glycine betaine is important for controlling the water content in the microbial cytoplasm [65], up-regulation of various genes for ABC-type proline/glycine betaine transport systems (ATPases and permeases) (COG1125, COG1174, COG4175, COG4176) suggests that the rhizobial betaine is also transported to the host cells under salt stress. Microbial betaine has been shown to significantly improve rice and tobacco salinity stress tolerance [66]. Furthermore, degradation of branched chain amino acids enhanced drought tolerance in *Arabidopsis thaliana* since they are used as substrates for the citric acid cycle under water stress [67]. In this study, transcripts for the degradation of all three branched chain amino acids were elevated in the nodules of plants treated with NaCl during RNA-Seq. This observation was also noted on quantifying the amount of valine, leucine, and isoleucine from nodules of salt treated plants were compared with those from untreated plants (Figure 5). Since genes for both ABC-type branched-chain amino acid transporting ATPases and permeases (COG0410, COG0411, COG0683, COG4177) were upregulated, elevated transcript levels for these transporters together with those for enzymes involved in the synthesis of leucine from oxoisovalerate (K00052), as well as isoleucine and valine from threonine and pyruvate (K17989, K00052) point to the importance of microbe-derived branched-chain amino acids for salt tolerance of the plants. Finally, transcripts for permeases (COG1176, COG1177) and the ATPase COG3842 involved in putrescene transport were elevated under salt stress. The importance of putrescene in the management of salt stress has been demonstrated before: exogenously applied phytoputrescene increased the efficiency of photosystem II, inhibited Na^+^ and Cl^−^ but increased K^+^ and Mg^2+^ uptake in different tissues of cucumber seedlings under salt stress [68]. Therefore, biosynthesis and transport of microbial putrescine to the host plants participates in the general plant tolerance response to salinity. Transcripts for cysteine synthase (COG0031) and the cysteine transport system K02424 were also upregulated under salt stress. Cysteine is a thiol amino acid that regulates plant growth and various abiotic stresses. The amino acid enhances the rate of photosynthesis, increased the quantity of proline, nitrates, phosphorus, and potassium, increased the activity of catalase and superoxide dismutase in salinity-stressed soybean seedlings [69].

Glutamate metabolism seems to play a vital role in microbe-induced salt tolerance in the host plant. Glutamate is potentially further metabolized to glutathione through elevated transcript levels for the glutathione synthase (K01920), to gamma aminobutyric acid through elevation of glutamate decarboxylase 1 (K01580), and to proline through elevated gamma-glutamyl phosphate reductase mRNA levels (COG0014). Furthermore, the bacterial cells in the nodule appear to promote glutamate biosynthesis for the formation of the described molecules, e.g., by elevating transcript levels for the biosynthesis of glutamate from 5-oxoproline (K01469) and from aspartate via the aspartate aminotransferase (K14454, COG0075). Glutathione can be transported across rhizobia membrane as indicated by higher transcript levels for the K13889, a substrate-binding protein of the glutathione transport system. Additionally, the level of GABA which is transported through the branched-chain amino acid transport permease K01999 [70] is upregulated under salt stress at the mRNA level. Since we did not observe the upregulation of any proline transporter or catabolic enzymes, this amino acid might be needed by the bacterium itself (Figure 6).

Both GABA and glutathione are important for plants under stress. Exogenously applied GABA increased soluble sugar and proline contents, alleviated damage to membranes, reduced oxidative damage, and increased enzymatic antioxidant activity of the salt-treated maize seedlings [71]. GABA increased plant height, dry and fresh weights, as well as the chlorophyll content in salt-stressed plants, while it reduced the Na^+^ concentration in the leaves and roots by preventing Na^+^ influx in roots and its transport to leaves in salt-stressed tomato [72]. On the other hand, glutathione also improves plant performance under salt stress: for instance, transgenic tobacco plants overproducing glutathione grew faster than the control seedlings under salt stress [73]. Glutathione protects the plasma membrane by prevention of lipid peroxidation and reduction of passive Na^+^ influx [74]. Salinity results in the accumulation of methylglyoxal which inhibits cell proliferation, increases protein degradation, forms adducts with guanyl nucleotide in DNA, and inactivates antioxidant defense system in plants [75]. Glutathione acts as a cofactor in the glyoxalase system that detoxifies methylglyoxal [76]. Quantitative real-time PCR indeed confirmed elevated transcript levels for the aspartate amino transferase, glutathione synthetase, and glutamate decarboxylase in the bacteroids during salinity stress, although only the expression of the glutathione synthetase gene was significantly different from that in the untreated cells at *p* ≤ 0.05 (Figure 7A). Furthermore, more GABA (significantly higher at *p* ≤ 0.05) and proline (not significant) accumulated in the nodules of salt-treated plants when compared to their untreated controls (Figure 7B).

#### 3.4.2. Inorganic Ion Transport and Metabolism

The importance of nutrient, ion and metabolite exchange between the two symbionts under salt stress is clearly visible by the large number of transporter genes which are upregulated in the rhizobium. Thus includes ABC-type transporters for phosphate (COG0573, COG0581, COG1117, COG3221, COG3638, COG3639), transporter for nitrate (COG0600), iron (COG1178), potassium (COG2060, COG2072, COG2216), manganese/zinc (COG1121), taurine (COG4521, COG4525), hemin (COG4559), enterochelin (COG4604, COG4605, COG4606), and siderophores (COG0609, COG1120, COG4615). Moreover, genes for non-ABC transporters such as for an alkaline phosphatase (COG1785), enterochelin esterase (COG2382), and nitrous oxidase (COG3420) were also upregulated. Salinity stress causes low nutrient ion levels, for example shortage of macronutrients (N, P, Mg, K and Ca) and micronutrients (Zn, Fe, Mn, and Cu) as a result of extreme imbalances in the uptake ratios of Na^+^/K^+^, Na^+^/Ca^2+^, and Cl^−^/NO_3_^−^ [77]. This increases the plant’s susceptibility to osmotic stress, ion injury, nutritional disorders, poor plant health and reduced yield [78].

The alkaline phosphatase (COG1785) removes phosphate groups from proteins, nucleotides, and alkaloids [79]. Up-regulation of the gene for this enzyme in bacteroids, as well as of genes for various extracellular phosphate transporting proteins indicates that dephosphorylation reactions in bacteria lead to accumulation of orthophosphates that are then exported to plant tissues during salt stress. Up-regulation of two genes for proteins involved in K^+^ transport (K^+^ transporting ATPase and a flavoprotein involved in K^+^ transport) likely enables efflux of microbial K^+^ ions into the host plant under salt stress. Elevated K^+^ levels in the salt-stressed plant improves plant growth, photosynthesis, antioxidant enzyme activities, and the ascorbate and glutathione contents, which results in lower oxidative stress as shown for the leaves of *Brassica campestris* [80]. Besides K^+^, NO_3_^−^ appears to be exported to the host since the microbial NO_3_^−^ transport system is upregulated as well. Nitrogen acquisition by the host plant may represent a potential strategy of overcoming salinity-mediated stress, since exogenously supplied nitrates decreased lipid peroxidation, electrolyte leakage from the leaves, and stimulated the antioxidant defense system, enhanced the osmolyte levels including free amino acids and soluble sugars in *Gossypium hirsutum* seedlings [81,82]. The identified ABC transporter for K^+^ confirms this hypothesis (Figure 8).

Since Fe exists in two different oxidation states (Fe^2+^ and Fe^3+^), it is a crucial component of redox reaction in respiration and photosynthesis [83]. Genes involved in Fe transport from nodule bacteroids to the host plant cells code for hemin, siderophore, and enterochelin biosynthesis enzymes and are upregulated under salt stress. It has been hypothesized that upon entry into the plant tissues, these molecules are further transported to the rhizosphere through plant sidorophore efflux mechanisms [84,85] to enhance Fe absorption from the soil by the salinity stressed plants. Furthermore, in addition to Fe, siderophores produced by *Bacillus subtilis*, *Arthrobacter sulfonivorans*, and *Enterococcus hirae* increased zinc mobilization in wheat genotypes [86].

Salinity limits water absorption in plants, and may lead to osmotic stress. Soluble carbohydrates or sugars play a crucial role as osmolytes in salinity-stressed plants by regulating osmotic adjustment and carbon storage in plants [87]. Several upregulated genes for permeases belonging to the major facilitator superfamily (COG0477) are possibly involved in the transport of simple sugars, inositols, amino acids, organophosphate esters, nucleosides, and Krebs cycle metabolites [88], or ABC-type sugar transporting permeases and ATPases (COG1129, COG1175, COG1653, COG1879, and COG3839) may facilitate the transport of maltose, galactose, raffinose, sorbitol, mannitol, or trehalose. Furthermore, genes for an arabinose efflux permease (COG2814) and citrate lyase (COG230) were also upregulated. In a study on the metabolite profiles of salt-stressed barley genotypes, sucrose, mannitol, inositol, and trehalose accumulated in roots while raffinose accumulated in both leaves and roots during salt stress [89]. Maltose accumulated in salt stressed *A. thaliana* [90] and might be involved in the biosynthesis of stress-dependent cell wall reorganization. Elevated maltose levels in the plastids of *A. thaliana* during salt stress might protect photosynthetic thylakoid proteins [91,92]. Up-regulation of soluble carbohydrate transporter genes therefore ensures that osmolytes are transported into the host plant during salinity stress. Citrate lyase is involved in the generation of acetyl-CoA and oxaloacetate which enters the citric acid cycle and may be converted to glutamate during salt stress. Maintaining cell wall integrity is critical for salinity tolerance. Arabinose is the integral component of many cell wall-localized glycoproteins and several cell wall polysaccharides [93]. Zhao et al. [94] reported reduced root elongation phenotype in an *A. thaliana* mutant incapable of arabinose biosynthesis during salt stress. The short root phenotype was rescued after the exogenous application of arabinose. Efflux of arabinose from bacteria to the host plant may also be one of the mechanisms that participate in salt tolerance resistance in rhizobia-colonized hosts.

## 4. Conclusions

We demonstrated that salt tolerant rhizobia can help host plants to cope with salinity stress. Salinity deprives the plant with water and nutrients through intolerable osmolarity and interreference in the absorption of important minerals. Expression profiles indicated that nodule bacteroids may restore the nutritional imbalance by mobilizing some of the minerals such as phosphates, calcium, potassium, iron, and nitrate into the stressed plants. Furthermore, rhizobia can synthesize agents such as siderophore and enterochelin and thus promote iron mobilization to improve the plants’ ability to absorb the ion. Upregulation of genes responsible for biosynthesis and transport of simple sugars, inositols, amino acids, organophosphate esters, nucleosides, and Krebs cycle metabolites is crucial in enhancing osmotolerance in saline stressed plants. Moreover, rhizobia supplied GABA and glutathione to the host plants prevents lipid peroxidation and activates antioxidant enzymes hence alleviating effects of oxidative stress in salt-stressed plants. Therefore, the ability of rhizobia to enhance salt tolerance in host plants is an important criterion for agricultural application.

## Figures and Tables

**Figure 1 cells-11-03628-f001:**
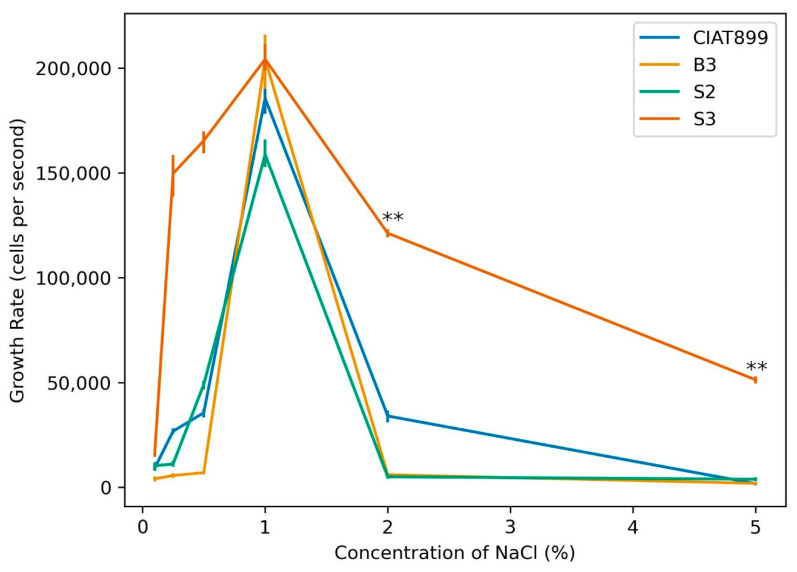
Growth response of rhizobia isolates B3, S2, and S3 in media with different NaCl concentrations in comparison to the commercial strain CIAT899 (one-way ANOVA; ** indicates *p* ≤ 0.001).

**Figure 2 cells-11-03628-f002:**
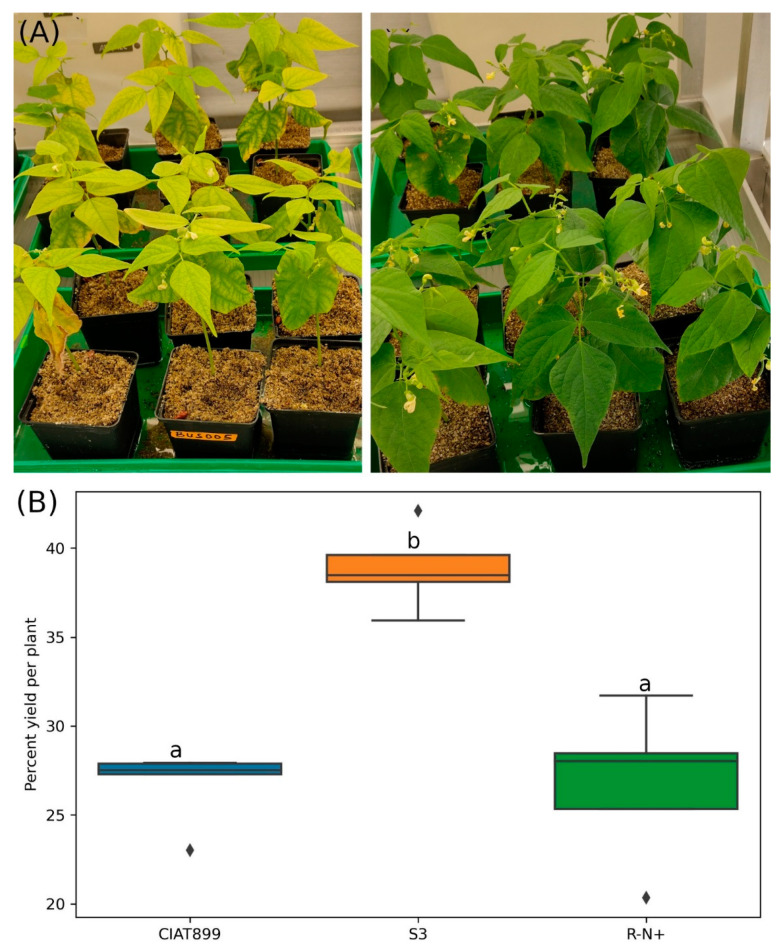
Common bean plants treated with 300 mM NaCl; (**A**; left panel) were not inoculated while (**A**) right panel) were inoculated with isolate S3, (**B**) Yields [(shoot dry weight of salt-treated plants/shoot dry weight of untreated control plants) × 100]. CIAT899: plants inoculated with *R. tropici* CIAT899, S3: plants inoculated with isolate S3, and R-N+: uninoculated plants supplied with nitrogen. The horizontal lines inside the boxes show the median percent yields, the diamonds (⧫) represent outlier values while significantly different yield percentages are indicated by different letters above the boxes (one-way ANOVA).

**Figure 3 cells-11-03628-f003:**
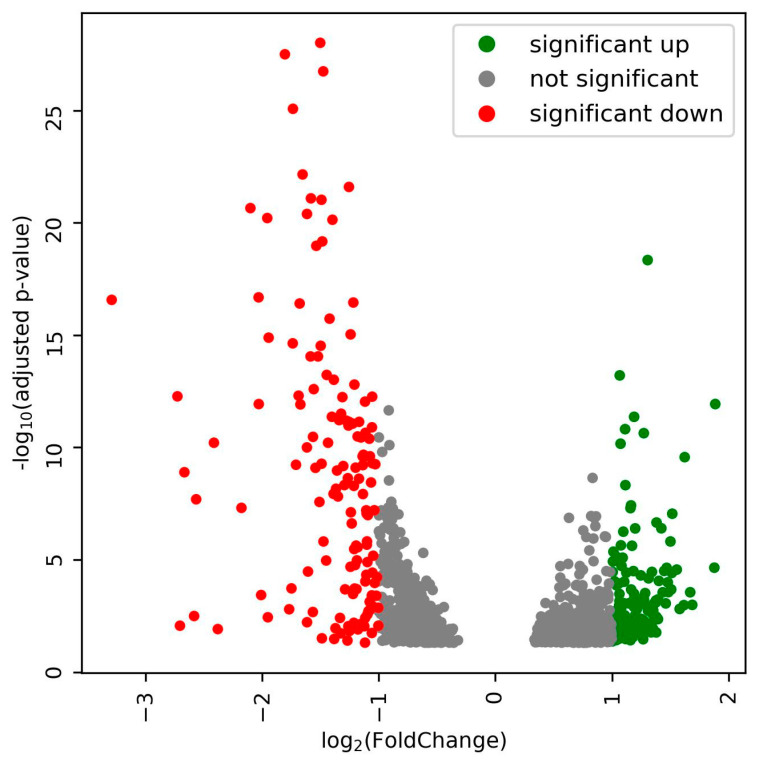
Volcano plot showing differentially expression genes in rhizobium bacteroids in the nodules when the infected plants were exposed to 300 mM NaCl.

**Figure 4 cells-11-03628-f004:**
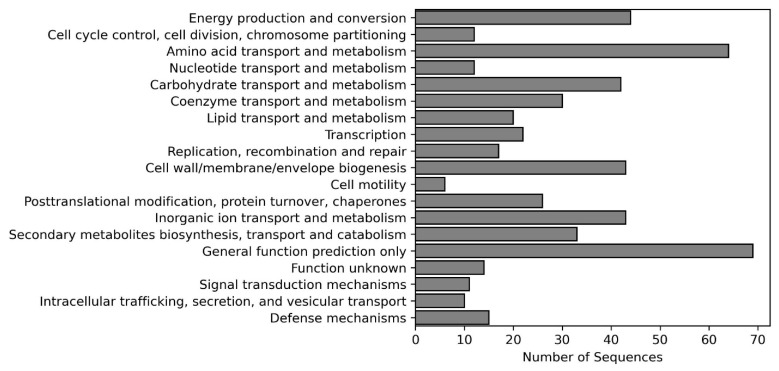
Functional analysis of upregulated genes after mapping to the database with cluster of orthologous groups (COGs).

**Figure 5 cells-11-03628-f005:**
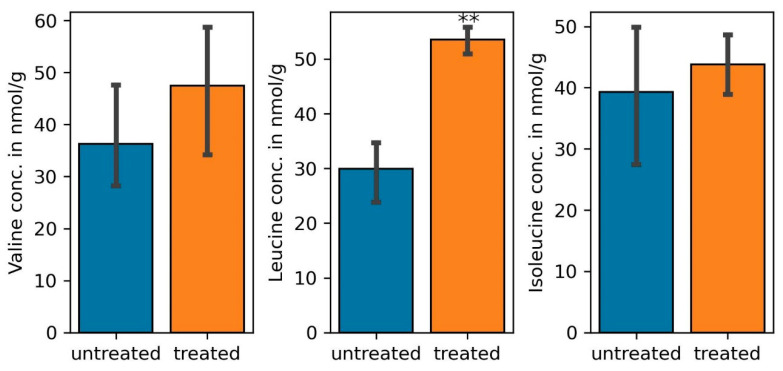
Quantity of branched chain amino acids, valine, leucine, and isoleucine in common bean nodules treated with 300 mM NaCl (treated) compared to those not treated with the salt (untreated). (one-way ANOVA; ** indicates *p* ≤ 0.001).

**Figure 6 cells-11-03628-f006:**
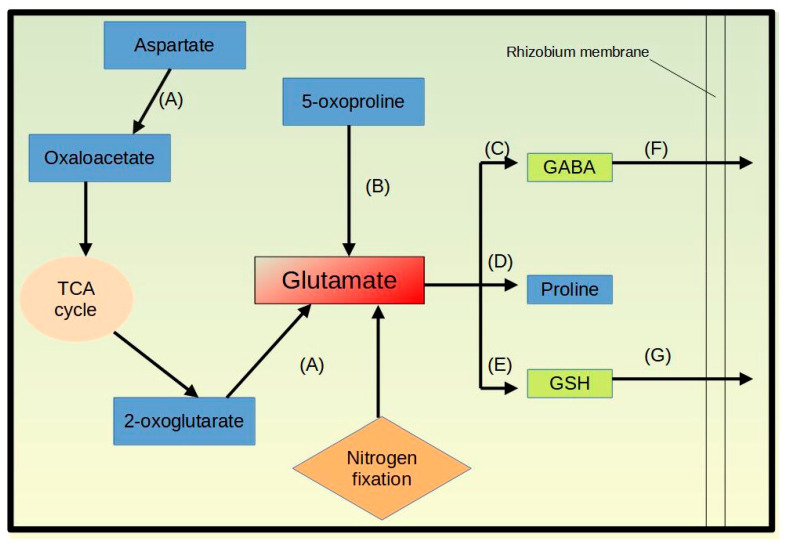
A schematic flow showing glutamate metabolic pathways in bacteroids under salinity stress. The letters represent the enzymes (**A**) aspartate amino transferase (K14454, COG0075), (**B**) 5-oxoprolinase (K01469), (**C**) glutamate decarboxylase 1 (K01580), (**D**) gamma-glutamyl phosphate reductase (COG0014), (**E**) glutathione synthase (K01920), (**F**) branched-chain amino acid transport system substrate-binding protein (K01999), and (**G**) glutathione transport system substrate-binding protein (K13889). This figure was drawn in Libre Office Draw.

**Figure 7 cells-11-03628-f007:**
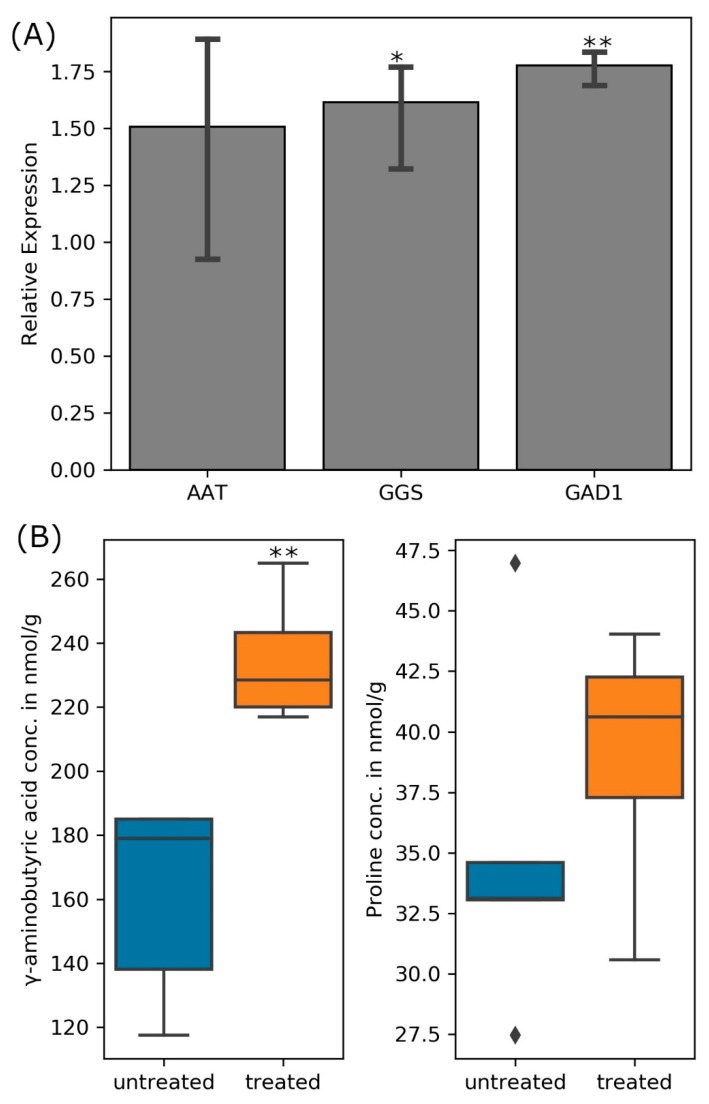
(**A**) Transcript abundance for nodule bacteroids’ aspartate aminotransferase (AAT), glutathione synthase (GGS), and glutamate decarboxylase (GAD1) when infected plants were treated 300 mM NaCl relative to untreated nodules and (**B**) quantity of GABA and proline in the nodules of treated and untreated (control) plants. (one-way ANOVA; * indicates *p* ≤ 0.05 while ** indicates *p* ≤ 0.001).

**Figure 8 cells-11-03628-f008:**
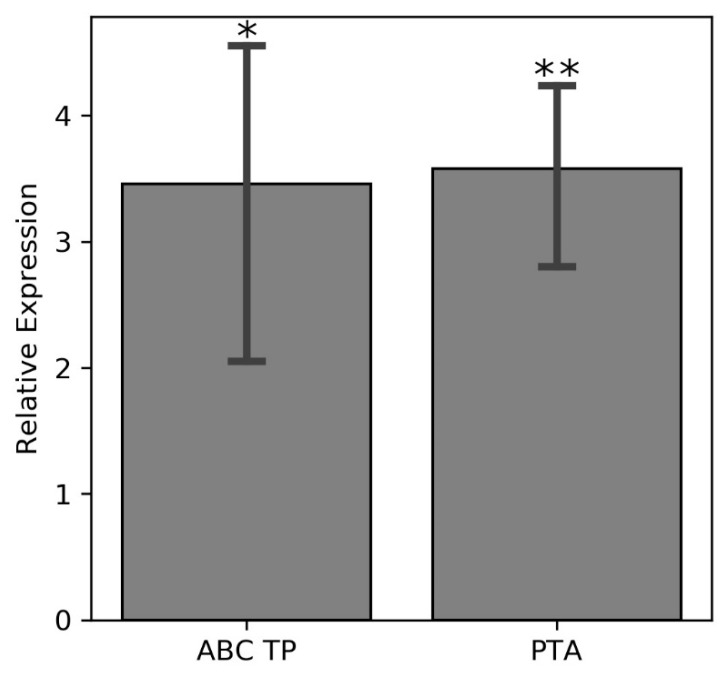
Relative expression levels of the genes for the ABC transporter permease and K^+^ transporter ATPase (COG2060) in the nodules of common beans with qPCR after treatment with 300 mM NaCl treated and untreated (control) plants. (one-way ANOVA; * indicates *p* ≤ 0.05 while ** indicates *p* ≤ 0.001).

**Table 1 cells-11-03628-t001:** Details on the analysis of GABA and amino acids proline, valine, leucine and isoleucine by LC-MS/MS [HPLC 1260 (Agilent Technologies)-QTRAP6500 (SCIEX)] in a positive ionization mode.

Compound	Q1	Q3	RT (min)	Internal Standard	IS Q1	IS Q3	DP	CE
GABA	104.1	87.1	0.5	^13^C,^15^N-Ala				
Proline	116.1	70.0	0.5	^13^C,^15^N-Pro	122.1	75.0	20	19
Valine	118.1	72.2	0.5	^13^C,^15^N-Val	124.1	77.2	20	13
Leucine	132.2	86.1	1.3	^13^C,^15^N-Leu	139.2	92.1	20	13
Isoleucine	132.2	86.1	1.1	^13^C,^15^N-Ile	139.2	92.1	20	13

## Data Availability

https://www.ncbi.nlm.nih.gov/geo/query/acc.cgi?&acc=GSE216374, accessed on 19 October 2022.

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
