# Peer review of "Rhizobia Contribute to Salinity Tolerance in Common Beans (Phaseolus vulgaris L.)"

_cells, 2022, doi:10.3390/cells11223628_

Round 1

Reviewer 1 Report

Major comment:

A supplementary Table with all the 1069 bacteroid gene differentially expressed in the nodules of 300 mM NaCl treated plants should be included, showing the number of reads and the statistical analysis. The authors show only a list with the COG number and the names of the differentially expressed genes. It is not enough. A complete list containing the data obtained in transcriptome analysis should be shown.

Minor comments

1.     Line 112: please change "ix" to "fix" nitrogen.

2.     Lines 255-256: The data described here are not shown in the MS. Please introduce a figure with them or add “data not shown” at the end of the sentence.

3.     Figure 2. Please change “I” and “II” to A and B, respectively. At Figure 2A, you can refer to I mages as right and left panels…

Insert in the legend which the statistical analysis was used and

Describe the median yield percentage graphical representations.

What does the diamonds represent?

Please describe at line 276 the treatment in their appearing order.

4.     Sentence of line 283-284 is not clear? What the meaning of this statement? Any statistical test was applied to these data comparing number of induced x repressed genes?

5.     Line  289, figure 3. It is no clear if the plants and symbionts were exposed to 300 mM NaCl. If yes, the information must be in the legend.

6.     Lines 310-313: Important!! Authors state that transcripts were elevated, however, figure 5 shows amino acid contents.

7.      Figure 7: what the relative expression is comparing? This should be in the legend.

8.     Line 441: Add a “.” at the end of the sentence.

9.     Line 442: Remove the “.” before the references.

Reviewer 2 Report

In this work, the authors analyzed the response to salt stress of the Rhizobium phaseoli S3 native strain during its symbiosis with common bean plants. Plants inoculated with this strain achieved significantly better than the other strains tested, then they selected the S3 strain to study its global expression profile under high NaCl concentrations.

Since salinity is one of the major challenges to global food security, the search for strains of rhizobia tolerant to salinity may be an alternative to establish symbiosis in conditions in which crop growth can be successful. Therefore, the results of this work are important for the area. The paper is interesting and reasonably well-written.

My principal concern is that it is difficult to conclude a great deal about the genes' specific role or a possible molecular mechanism.

It is known that salinity negatively affects the Rhizobium-legume symbiosis. High salt concentration interferes with the establishment and development of symbiotic nodules. This negative effect results in inefficient nodules with reduced leghemoglobin and nitrogenase activity. Not all readers may know about nodule functionality; this should be a bit more detailed in the introduction.

In the same regard, why do the authors not report any data concerning the symbiotic efficiency or nodule functionality (leghemoglobin content and ARA) of the S3 strain?

Is the S3 strain can fix nitrogen?

Is the S3 strain able to form nodules when the seedlings are inoculated in the presence of this high NaCl concentration (at the beginning of the symbiosis)?

Why were the antioxidant enzymes such as catalase or superoxide dismutase not differentially expressed? It has been reported that these enzymes are part of the bacterial response to salt stress.

Information about the primers used for RNA-Seq validation and details of the RNA-Seq analysis must be presented as supplementary material.

Which reference gene was used for normalized the expression levels of genes tested?

The relative expression was determined as 2DDCT?

Line 112. Change ix by fix

Line 163. 260/280 nm of….and 260/230 nm of…. Optical, COMPLETE

Line 237 delete at

Genes upregulated were 146 (line 284) or 145 (line 292)

Lines 320 and 321, change phytoputrescene and putrescene by phytoputrescine and putrescine, respectively

Round 2

Reviewer 1 Report

MS is ok in the present form.